# A Systematic Review of *Curtisia dentata* Endemic to South Africa: Phytochemistry, Pharmacology, and Toxicology

**DOI:** 10.3390/life13112159

**Published:** 2023-11-03

**Authors:** Maropeng Vellry Raletsena, Ofentse Jacob Pooe, Nkoana Ishmael Mongalo

**Affiliations:** 1College of Agriculture and Environmental Sciences Laboratories, University of South Africa, Private Bag X06, Florida 0610, South Africa; mongani@unisa.ac.za; 2Department of Biochemistry, School of Life Sciences, University of KwaZulu Natal, Private Bag X54001, Durban 4000, South Africa; pooeo@ukzn.ac.za

**Keywords:** ethno-medicinal uses, pharmacological activity, phytochemistry, triterpenes, cytotoxicity, traditional medicine, Southern Africa

## Abstract

The use of traditional medicine in treating a variety of both human and animal infections is ancient and still relevant. This is due to the resistance exhibited by most pathogenic microbial stains to currently-used antibiotics. The current work reports the phytochemistry, ethno-medicinal uses, toxicology, and most important pharmacological activities that validate the use of the plant species in African traditional medicine. *Curtisia dendata* is used in the treatment of many human and animal infections, including diarrhea, skin and related conditions, sexually transmitted infections, cancer, and a variety of ethno-veterinary infections. Pharmacologically, the plant species exhibited potent antimicrobial activity against a variety of pathogens. Further, both extracts and compounds isolated from the plant species exhibited potent antioxidant, anticancer, anti-parasitic, anti-inflammatory, and other important biological activities. Phytochemically, the plant species possess a variety of compounds, particularly triterpenes, that may well explain the various pharmacological activities of the plant species. The toxicological parameters, antimicrobial activities against microorganisms related to sexually transmitted infections, anti-diabetic effects, and inflammatory properties of the plant species are not well studied and still need to be explored. The biological activities observed validate the use of the plant species in African traditional medicine, particularly in the treatment of pulmonary infections associated with *Mycobacterium* species, and may well be due to the presence of triterpenes prevalent in the leaves.

## 1. Introduction

South Africa is blessed with a rich plant biodiversity, which is mostly used as a first line of defense against a variety of both human and animal infections. Human infections are devastating and are further aggravated by the opportunistic infections associated with HIV/AIDS, cancer, the resistance of microorganisms to commonly available and used antibiotics, the occurrence of mycobacterium infections, and a variety of outbreaks [1,2]. The plant species used for medicinal purposes possess a variety of phytochemicals and secondary metabolites that yield various important pharmacological activities, which may include anticancer, antimicrobial, antioxidant, anti-inflammatory, antiviral, anti-parasitic, anti-diabetic, insecticidal, anti-molluscidal, anti-pesticide, wound healing effect, and anti-malarial activities [3,4,5]. The use of these plants as a source of medicine is further advantaged by their ever-present presence throughout the year, production of no side effects, and belief that they surpass most of the microbial resistance [6]. However, the safety profiles of such plant species are of paramount importance and need to be explored, as some medicinal plants exhibit some toxicity, both in vivo and in vitro, at low concentrations and may eventually result in death [7].

*Curtisia* is a genus that comprises only one plant species, *Curtisia dendata* (Burm.f.) C.A. Sm., in Southern Africa. *Curtisia* species generally comprise a fruiting body characterized by isodiametric sclereids, a single seed per locule, a highly visible axial vascular canal, apical placentation, and at least four germination valves [8]. Currently, the comprehensive indigenous medicinal uses, phytochemistry, and pharmacological activity of *Curtisia dentata*, an indigenous multipurpose and threatened Southern African plant species, are reported. Most of the reported pharmacological activity is reported from the leaves, as the endemic plant is mostly available in protected areas. Earlier, some antimicrobial and ethnomedicinal uses of the plant species were reported [9].

## 2. Materials and Methods

### Collection of Data

The information reported in the current paper was collected from a literature search using various computerized databases such as ScienceDirect, Scopus, Scielo, Scifinder, PubMed, Web of Science, and Google Scholar. Additional information was further retrieved from various academic dissertations, theses, and general plant sciences, ethnomedicine, and other relevant ethnobotanical books. This was done following the guidelines in the Preferred Reporting Items for Systematic Reviews and Meta-Analyses (PRISMA) statement [10]. Key words such as *Curtisia dentata*, ethnomedicinal uses, survey, ethnopharmacological aspects, biological activity, antimicrobial activity, anti-bacterial, anti-fungal, pharmacological properties, biological activity, cytotoxicity, toxicology, anti-cancer, phytochemical components, nutritional composition, phenolic compounds, polyphenols, phytochemistry, anti-inflammatory, antioxidant properties, anti-diabetic, anti-malarial, pesticidal effect, anti-parasitic, anthelmintic, anti-convulsant, and insecticidal effect. The data were collected with help from library staff at the University of South Africa (Florida Campus). The summary of the sources used in the current work is shown in Figure 1 below.

## 3. Taxonomy, Botany, Conservation Status, and Distribution of *C. dentata*

*Curtisia dentata* belongs to the family Curtisiaceae, formerly Cornaceae, and the order Cornales. The family is named after William Curtis, the founder of Curtis’s Botanical Magazine. *Curtisia dentata* (Figure 2) is a medium- to large-sized tree with dark green, simple, oblong, toothed, broadly based, sharply pointed, and oppositely hairy leaves that occur in pairs on the twigs [11]. *C. dendata* flowers are creamy, odorless, self-pollinated, and occur in clusters [12]. The petals are twice as long as the calyx tube and oblong in shape, while the stamens are equal in length to the petals [13]. The fruits, usually referred to as drupes, are about 10 mm in diameter, fleshy, red when ripe, and edible [14]. The stem bark is brownish-grey, hard, and may become fissured as it ages [15]. The plant species is endemic to Southern Africa and is found in South Africa, from the Western Cape Province extending along the coast down to Limpopo Province, Eastern Zimbabwe [16]. The plant species is also found in Mozambique and some parts of Swaziland and Mozambique. Elsewhere, the plant species occurs in Italy and Brazil [17].

Currently, the plant species is threatened and is found mostly in conserved areas, which include parks, forests, and botanical gardens in Southern Africa [18,19,20,21,22,23,24,25,26,27]. Although the plant species is still available in some parts of South Africa, it is highly threatened by bark stripping for medicinal purposes [28]. The stem bark is harvested in bulk quantities and traded in most cities and traditional medicine markets throughout South Africa [29,30,31,32,33,34,35]. However, it is important to notice that the plant species has also declined drastically in the wild, not only due to excessive harvesting but also due to the expansion of human informal settlements that are rapidly increasing [36]. According to Van Rensburg [37], the “South African Green Heritage Campaign” supported and recommended that the indigenous tree should be planted for ornament, shade, and shelter by various schools, governmental departments, and many youth organizations in the future as a conservation measure.

## 4. Phytochemistry

The acetone extract from the leaves exhibited a saponin content of 3.33 mg/mLGAE, a total phenolic content of 125.12 mg/gGAE, and a total flavonoid content of 27.69 mg/GQE [38,39]. Both the ethanol and aqueous extracts from the stem exhibited reasonable amounts of phytoconstituents, including steroids, saponins, tannins, glycosides, and phenols, in standard tests. Elsewhere, the 70% ethanol extract of the stem bark exhibited total phenols, total flavonoids, saponins, tannins, alkaloids, and steroids at 14.86 mg/g, 13.64 mg/g, 13.26%, 0.51%, 0.51%, and 1.42%, respectively [40]. The phytochemistry of the plant species is well reported in the literature using both NMR characterization [41,42,43,44,45,46,47,48] and gas-chromatography-mass spectrometry (GC-MS) analysis [49]. Figure 2 depicts various compounds identified in *C. dentata*. The leaves of the plant species revealed the presence of a variety of compounds, mostly dominated by triterpenes and steroids (Figure 2). Compounds such as lupeol **1**, betulinic acid **2**, ursolic acid **3,** and 2α-hydroxyursolic acid **4** were isolated from leaves [46]. The other important triterpenes include α-amyrin **5**, β-amyrin **6**, β-sitosterol **7,** and Vitamin E **8** [43,44]. The other compounds identified by GC-MS with % area >1 include catechol **9**, 1,6-anhydro-β-D-Gloucopyranose **10,** neophytadiene **11**, palmitic acid **12**, 1-docosene **13**, 4-propylcatechol **14** [50]. The presence of these phytocompounds in higher quantities may well explain the biological activity observed and other important pharmacological properties, both in vitro and in vivo. For example, betulin and betulinic acid are well known for their anti-inflammatory, antibacterial, antiviral, antidiabetic, antimalarial, anti-HIV, and antitumor effects [51,52,53,54,55]. These diverse biological activities and wide distribution within the kingdom of Plantae at large validate the use of traditional medicine holistically in the treatment of many human and animal infections worldwide. Van Wyk and Prinsloo [48] evaluated both the seasonal and regional metabolite profiling of the stem bark using NMR. The clustering of the samples, both hydrophilic and lipophilic metabolite profiles, resulted in varying changes seasonally. These changes were explained by differences in terms of climatic patterns and environmental conditions. This may include seasonal differences in the amount of rainfall, water and nutrient availability, and abiotic stress factors such as temperature, UV exposure, photoperiod, drought, and flooding for the specific region.

## 5. Pharmacological Activities of *Curtisia dentata* Extracts and Isolated Compounds

### 5.1. Anti-Microbial Activity

Virulent microorganisms develop some resistance to common antibiotics, mostly used in developing and underdeveloped countries worldwide. The resistance may be due to the development of genes that resist the entry of different drugs into the microbial cell, thereby allowing pathogenic microbes to grow and resulting in devastating effects even if a potent drug is administered to a patient. According to Brown and Wright [56], from 2000 until now, the discovery of antibiotics has been lower compared to the discovery of antibiotic-resistant microorganisms. Some authors propose the use of antibiotics in conjunction with medicinal plant extracts as an alternative to curb problems associated with microbial resistance [57]. These resistances warrant a thoroughly comprehensive search for new antibiotics, mostly of plant origin, to ease pressure on currently available drugs in various pharmaceutical setups worldwide. It is important to note that even though the antimicrobial activity of various plant species has been well studied, the benchmark for such activities differs from one author to the next [58,59]. In the current work, plant extracts, and isolated compounds with MIC values of less than or equal to 0.1 mg/mL are referred to as active and have potential and noteworthy antimicrobial activity, while 0.63 mg/mL MIC 1.0 mg·mL has moderate antimicrobial activity [60,61,62]. The MIC > 1 mg/mL is therefore referred to as less active or inactive. Using these as a standard benchmark, it is clear that the extracts, fractions, isolated compounds, and derivatives from *Curtisia dendata* exhibited noteworthy antimicrobial activity against microorganisms that include *Microsporum canis*, *Sporothrix schenkii*, *Candida albicans*, *Cryptococcus neoformans*, *Mycobacterium tuberculosis*, *Escherichia coli*, *Bacillus cereus*, *Bacillus subtilis*, *Shigella sonnei*, *Shigella typhimurium*, *Streptococcus pyogenes*, *Enterococcus faecalis*, *Pseudomonas aeruginosa*, *Mycoplasma hominis*, *Staphylococcus aureus,* and *Acinetobacter lwoffii* [41,42,43,44,45,46,47,49,63,64,65]. These results, in a way, validate the use of the plant species in the treatment of a variety of infections, particularly tuberculosis, sexually transmitted infections, diarrhea, opportunistic infections associated with HIV/AIDS, and skin-related infections, as cited in the literature for such uses [2,66,67]. However, it is of paramount importance to explore the antimicrobial activity of such medicinal plants and isolated compounds from the plant species in vivo, as in vitro results may not always translate into in vivo studies.

Acetone extract from *Curtisia dendata* leaf exhibited notable MIC values of 0.02 and 0.08 mg/mL against *Microsporum canis* and *Sporothrix schenkii,* respectively, compared to other extracts [47,57,68]. Further, hexane extract revealed a MIC value of 0.30 mg/mL against both *Cryptococcus neoformans* and *Microsporum canis*. Elsewhere, the methanol extracts from both stem bark and roots exhibited MIC values of 0.08 mg/mL against *Candida albicans*, while the acetone extract from the leaf exhibited MIC value of 0.01 mg/mL against a similar fungal strain isolated from an HIV/AIDS patient [41,42,43,44,45]. Elsewhere, betulinic acid isolated from the leaves exhibited a potent MIC value of 0.03 and 0.02 mg/mL against *C. neoformans* and *C. albicans,* respectively [65]. It is important to notice that both *M. schenkii* and *M. canis* are implicated in human and animal cutaneous skin infections and/or dermatophytosis [69,70,71]. *Candida albicans* is an important pathogen that may cause a variety of infections, including skin and lesions infections and opportunistic infections associated with HIV (oral candidiasis), while *Cryptococcus neoformans* could give rise to various hospital-acquired infections and may also cause central nervous system and kidney infections [72,73]. Clearly, the compounds and extracts from *C. dendata* exhibit potential antifungal activity against pathogenic fungal strains. This validates the use of these plants in the treatment of infections.

#### 5.1.1. Antifungal Activity

In other studies, a comprehensive antifungal study of the plant species against a variety of mycotoxigenic fungi have been carried out [48]. The acetone extract exhibited MIC values of 0.08 and 0.16 mg/mL against *Aspergillus ochraceous* and *Furasium verticilloides,* respectively, at both 24 and 48 h incubation periods. The extract further exhibited MIC values of 0.63 and 1.25 mg/mL against *Aspergillus flavus* at 24 and 48 h incubation periods, respectively. Although the extract exhibited potential antifungal activity against *A. achraceous*, there is still a need to explore the anti-mycotoxigenic effect of the plant species against other pathogens, which are well known to produce mycotoxins that have a negative impact on food security at large. In the mycelial growth inhibition study, the extract exhibited 32.79, 24.61, and 18.61% inhibition against *F. verticilloides* at 3, 6-, and 9-day incubation periods. These may well suggest that the mode of action of the plant extract against the pathogen may be through the inhibition of mycelial growth inhibition (MGI). These may also suggest that the mode of action of the plant species against other pathogens is not through MGI, and that still needs to be explored. It is evident that the use of the leaves from *C. dentata* may well inhibit the growth of various crop-infecting fungal strains, thereby promoting high yields in crops such as potato, maize, wheat, and other foodstuffs consumed as staple foods throughout the world. These may also assist in addressing matters relating to food security and hunger. However, it is important to consider exploring the impact of these plant extracts and isolated compounds on crops such as maize and wheat treated with higher concentrations of mycotoxigenic fungal strains. Furthermore, mycotoxins produced by such fungal strains need to be quantified.

#### 5.1.2. Anti-Mycobacterial Activity

Tuberculosis (TB) is an important cause of morbidity and mortality and is caused by the bacterium *Mycobacterium tuberculosis* [74]. Approximately 10 million people are infected each year, making TB one of the top 10 causes of death worldwide [75]. In fact, TB is the second largest killer world-wide and due to its resistance to variety of antibiotics, it has resulted in increase in search of plant-based sources as alternative antibiotics [76,77]. Both methanolic extracts from stem bark and leaf exhibited MIC values of 1.25 mg/mL against *Mycobacterium smegmatis* [49]. The antimycobacterial activity of fractions and isolated compounds from *Curtisia dendata* leaf ethanol extracts has been explored using the Microplate Alamar Blue Assay [44,78]. The methylene chloride and acetone fractions exhibited noteworthy MIC values of 0.022 and 0.044 mg/mL against *Mycobacterium tuberculosis*, while ursolic-acetate (and betulinic acid acetate derivatives synthesized from ursolic acid and betulinic acid isolated from *C. dendata* extract) exhibited MIC values of 0.003 and 0.02 mg/mL against *M. tuberculosis,* respectively, in a similar assay. It is important to note that betulinic acid (BA) and ursolic acid (UA) exhibited MIC values of >0.05 mg/mL against similar *Mycobacterium* species in a similar assay. These may well suggest that the compounds UA and BA may well be used as starting compounds that yield excellent anti-mycobacterial compounds that could serve as alternative medicines for the treatment of resistant *Mycobacterium* infections. These results agree with those of Akgün et al. [79], who reported that the derivatives from phthalimide exhibited noteworthy anti-mycobacterium activity compared to the compound itself. The result of the current work validates the use of the plant species, particularly the leaves, in the treatment of *Mycobacterium*-related infections.

#### 5.1.3. Antibacterial Activity

The antibacterial activity of hexane, water, and ethanol extracts from the stem bark of *Curtisa dentata* has been investigated using both disc diffusion (DD) and microdilution (MD) assays [80]. Only the ethanol extract exhibited a zone of inhibition of 0.28 mm and a minimum inhibitory concentration of 0.78 mg/mL against *Bacillus subtilis*. Wintola and Afolayan [81] investigated the antibacterial activity of aqueous, ethanol, and acetone stem bark extracts against a plethora of bacterial strains using both agar well diffusion (AWD) and MD assays. Although acetone extract exhibited the highest zone of inhibition of 25 mm against both *Shigella sonnei* and *Streptococcus pyogenes* at an unknown concentration, these results are not so important as the AWD and DD assays are not a good method to compare with other studies. According to Mongalo et al. [70], these methods are dependent on several factors, including the type of agar used, the incubation period, the size of the inoculum culture, the diffusion rate of the extract, and other conditions. In the MD assay, the acetone extracts yielded a MIC value of 0.01 mg/mL against *Escherichia coli*, *Shigella sonnei,* and *Streptococcus pyogenes*, while the aqueous and ethanol extracts exhibited a similar MIC value of 0.01 mg/mL against *Escherichia coli* and *Bacillus cereus,* respectively [81]. Furthermore, acetone and ethanol extracts exhibited a MIC value of 0.02 mg/mL against *Salmonella typhimurium*, *Enterococcus faecalis*, *Bacillus subtilis,* and *Shigella flexineri,* while aqueous extracts yielded a similar MIC value against *Streptococcus* aureus and *Bacillus subtilis*.

The antibacterial activity of the plant species, particularly the aqueous extract, against *Shigella sonnei*, *Shigella flexineri*, *Salmonella typhimurium,* and *Escherichia coli* may well validate the use of the plant species in the treatment of diarrhea and dysentery, as documented in the literature [46,47,63]. Recently, there have been reports that point out that *Salmonella*, *Escherichia,* and *Shigella* species are implicated in causing acute diarrhea, which may well result in morbidity due to their resistance to most antibiotics used in the current hospital setups worldwide [82,83,84,85]. According to Fukushima et al. [86], these bacterial strains exhibited closely related genes in the phylogenetic analysis using the 16S r RNA gene sequence. Elsewhere, the ethanolic extract from stem bark exhibited a MIC value of 0.1 mg/mL against *Acinetibacter lwoffii* isolated from wastewater systems [63].

Soyingbe et al. [2] investigated the antibacterial activity of methanol, ethyl acetate, acetone, and aqueous leaf extracts against a variety of human and animal-infecting bacterial strains. The extracts from *C. dendata* exhibited MIC values of 5 and >10 mg/mL against *Enterococcus faecalis,* while Afolayan and Wintola [81] reported an MIC value of 0.02 mg/mL from stem bark methanol and acetone extracts against *E. faecalis*. These may well explain why the active phytoconstituents from the plant species could be embedded in the stem bark rather than the leaves. In a comparison study, Shai et al. [68] showed that the antibacterial activity of the plant species is higher in the leaves and twigs compared to the stem bark. These shows the most active phytocompounds should be embedded in the leaves and more soluble in acetone. Shai et al. [46] investigated the antibacterial activity of hexane, dichloromethane, and acetone leaf extracts against *Enterococcus faecalis*, *Escherichia coli*, *Staphylococcus aureus,* and *Pseudomonas aeruginosa* using both bioautography and microdilution assays. The bioautograms developed in chloroform/ethyl acetate/formic acid (5:4:1) resulted in the various compounds showing activity against selected microorganisms. Notably, three compounds from acetone extracts exhibited some prominent inhibition of the selected microbes at Rf values of 0.36, 0.54, and 0.88. In the microdilution assay, acetone extract exhibited a MIC value of 0.05 mg/mL against *Enterococcus faecalis* and a notable MIC value of 0.80 mg/mL against both *Escherichia coli* and *Pseudomonas aeruginosa,* while hexane extract exhibited a MIC value of 0.60 mg/mL against *E. coli, P. aeruginosa,* and *E. faecalis*. Nielsen et al. [48] investigated the antibacterial activity of both stem bark and leaf methanol extracts against methicillin-resistant *Staphylococcus aureus*, carbenicillin-resistant *Pseudomonas aeruginosa*, and ampicillin-resistant *Klebsiella pneumoniae*. The leaves exhibited weaker antibacterial activity compared to the stem bark extracts, yielding MIC values ranging from 0.63 to 1.25 mg/mL against selected microbes, while stem bark extracts exhibited MIC values of 0.31 mg/mL against the three antibiotic-resistant strains.

The antibacterial activity of extracts and isolated compounds from *Curtisia dentata* leaves has been investigated [41,42,43,44]. Ethanol and chloroform extracts exhibited a notable MIC value of 0.10 mg/mL against *Mycoplasma hominis*, while betulinic acid yielded potent MIC values of 0.06 and 0.01 mg/mL against *Staphylococcus aureus* (isolated from HIV/AIDS patients) and *M. hominis*, respectively. Furthermore, ursolic acid revealed a potent MIC value of 0.05 mg/mL against *M. hominis*. These results support the use of plant species in the treatment of sexually transmitted infections and some of the opportunistic infections associated with HIV/AIDS. Recently, *Mycoplasma* microbes, along with *Ureaplasma* and *Gardnerella* species, are believed to be one of the wombs infecting microbes, resulting in infertility and other urogenital infections [87,88,89]. Isolated compounds from *Curtisia dendata* leaves exhibited noteworthy antibacterial activity against some important pathogens [46]. Ursolic acid (UA) and hydroxylursolic acid (HA) exhibited better antibacterial activity compared to lupeol and betulinic acid. HA yielded MIC values of 0.03, 0.008, and 0.04 mg/mL against *Staphylococcus aureus*, *Pseudomonas aeruginosa,* and *Enterococcus faecalis,* respectively. Further, UA exhibited MIC values of 0.004 and 0.05 mg/mL against *P. aeruginosa* and *E. faecalis* respectively. Although Southern African traditional medicine recommends the use of stem bark for the treatment of a variety of infections [9], it is important to note that the biological activity of the leaves and isolated compounds from leaves reported in the current work support the use of leaves as a substitute for stem bark as a conservation measure for plant species. Although the antibacterial activity is more pronounced in the stem bark, the activity of the leaves is of paramount importance.

### 5.2. Anti-Cancer Activity

The anti-cancer activity of the compounds isolated from *Curtisia dendata* methanol extract has been studied against HepG2 [42]. Although betulinic acid, ursolic acid, and β-sitosterol yielded a weaker LC_50_ value > 300 µg/mL, lupeol exhibited a LC_50_ value of 289.4 µg/mL against HepG2. These results may well explain why the isolated compounds did not yield any potential inhibition of HepG2. In other studies, conducted by Soyingbe et al. [2], various extracts from the leaves were evaluated for anti-cancer activity against MCF-7, Caco-2, HeLa, and A549 cell lines. The extracts inhibited the cell lines in the following order: acetone > ethyl acetate > methanol > water. Acetone extract exhibited notable LC_50_ values of 41.55, 43.24, 45.13, and 57.35 µg/mL against A547, MCF-7, HeLa, and Caco-2, respectively. According to the United States Cancer Institute, a plant extract is referred to as inhibitory to cancer cell lines if it exhibits an LC_50_ of less than 20 μg/mL [90]. The extracts with LC_50_ values ranging from >20 μg/mL to 45 μg/mL moderately exhibit some cytotoxic effect or anticancer activity against the tested cell line of choice [91]. These results may well suggest that the anticancer activity of the plant species is more pronounced in the acetone extract compared to methanol, ethyl acetate, and aqueous extracts. These results are worrying because traditional medicine uses water as a solvent. However, in some instances, particularly on the skin and related infections, the plant material is powdered and applied directly to the skin until the infection subsides or is completely healed. Further work is essential in determining the anticancer activity of the plant species against other cancerous cell lines. Furthermore, the cytotoxic effect of the stem bark extracts still needs to be explored for comparison purposes with the extracts and compounds isolated from the leaves.

### 5.3. Antioxidant Activity

Reactive Oxygen Species (ROS) and Reactive Nitrogen Species (RNS), including peroxides, hydroxyl radicals, superoxides, and nitrous oxide, are generated in living organisms through cellular metabolism and are known to play a major role in oxidative cellular damage and various stresses, resulting in a variety of diseases that are detrimental to both human and animal health [92]. Although free radicals are known to cause diseases, they can be quenched by the antioxidant abilities of various foodstuffs and many other important secondary metabolites from plant sources, particularly medicinal plants. The antioxidant activity of crude extracts and compounds isolated from *Curtisia dendata* is well documented [42,48,52]. The ethyl acetate and acetone leaf extracts were investigated for antioxidant activity against 2,2-diphenyl-1-picrylhydrazyl (DPPH) at concentrations ranging from 0.08 to 5.0 mg/100 mL [43]. At 0.60 mg/mL, the acetone and ethyl acetate extracts exhibited 41.4 and 14.0% inhibition of DPPH, respectively, while ascorbic acid (the control drug) revealed 100% inhibition of the stable free radical. These results may well suggest that the free radical-scavenging compounds from the leaves are more soluble in acetone than ethyl acetate. Further, these results corroborate those of other authors [93,94], who reported that acetone extracts yielded higher inhibition of free radicals compared to other solvents. However, it is important to note that other authors prefer the use of methanol as an extracting solvent for antioxidant assays [95,96,97]. However, the solubility of phytocompounds in a solvent might well depend on the type of phytocompounds, polarity of such compounds, time, and/or method of extraction used [61].

Elsewhere, the 70% ethanol extract from the stem bark was evaluated for antioxidant activity using DPPH, 2,2’-azino-bis(3-ethylbenzothiazoline-6-sulfonic acid (ABTS), nitric oxide (NO), hydrogen peroxide (H_2_O_2_), and lipid peroxidation (LPO) assays [40]. The extract exhibited IC_50_ values of 18.0 and 17.0 µg/mL against ABTS and DPPH free radicals, respectively. These results suggest that the extract inhibits both DPPH and ABTS equally. Contrarily, other authors reported that the plant extracts inhibit both free radicals unequally [98]. The extract further exhibited IC_50_ values of 52, 159, and 60 µg/mL in NO, H_2_O_2_, and LPO assays, respectively. Generally, the results suggest that the extract inhibits DPPH and ABTS much better than other radicals. Further, the extract is a good antioxidants, as it yielded notable activity in all five assays. A good antioxidant should possess notable activity in at least three different assays [99]. The acetone extract of the leaf exhibited an IC_50_ value of 22.57 µg/mL against DPPH [38]. Doughari et al. [100] evaluated the antioxidant activity of various stem bark, leaf, and root extracts using reducing power (RP) and DPPH assays at a concentration of 0.1 mg/mL. The distilled water, chloroform, and acetone extracts of the leaf exhibited weaker antioxidant activity, yielding 1.62, 3.22, and 1.07% inhibition, respectively. The extracts inhibited DPPH much better than RP, with the ethanol extract from the root bark exhibiting the highest inhibition of DPPH, yielding 62.43%. Furthermore, the correlation between antioxidant activity and phenolic contents was reported. These results validate the use of the plant species in the treatment and management of free radical-associated illnesses. It should be noted that plant species possessing high antioxidant activity are likely to alleviate the onset of highly reactive molecules that are capable of initiating, elongating, and resulting in various forms of disease.

### 5.4. Anti-Inflammatory Activity

The anti-inflammatory activity of *Curtisia dendata* has been reported [38]. The acetone extract of the leaf exhibited an IC_50_ value of 95.38 µg/mL, while the control drug revealed an IC_50_ value of 9.02 µg/mL in vitro in an A 5-lipoxygenase assay. Although these results are of vital importance, it is important to note that the preferred medicinal plants for treating inflammation and related symptoms should have a selective inhibition of COX-2. That is, the phyto-constituents of such medicinal plants’ species should inhibit COX-2 and not COX-1, as they are associated with the essential organs within the body [101]. Thereafter, the mode or mechanisms of action could be explored. The inflammatory activity of the plant species still needs to be explored, as it is reportedly used in the treatment of many important illnesses that could be associated with inflammation.

### 5.5. Anti-Parasitic and Anti-Amoebic Activity

Global food security will always require the production of more foodstuffs using various resources and land more efficiently. However, it is important to consider the role of parasites in agriculture as one of the most prominent threats to food security, production, and quality. The resistance of nematodes that infect ruminants is becoming more common, with some organizations incorporating the use of dual anthelmintic compounds with a broad spectrum and different mechanisms of action on helminths with the intention of reducing resistance and the treatment regime [102]. The anti-parasitic activity of leaf extracts and isolated compounds from *Curtisia dendata* against *Haemonchus contortus*, *Trichostrongylus colubriformis,* and *Caenorhabditis elegans* has been studied [103]. Both acetone and dichloromethane extracts exhibited a notable inhibition of all the parasites at the lowest concentration of 160 µg/mL. Betulinic acid and lupeol exhibited inhibition of the parasites at 1000 and 200 µg/mL, respectively. *Haemonchus contortus* is one of the gastrointestinal parasites that infects sheep and goats and has a devastating effect on meat production, thereby resulting in negative financial effects on breeders [104]. The results suggest that the active extracts may be due to the synergistic effect of various compounds other than betulinic acid and further validate the use of the plant species in the management and treatment of ethno-veterinary infections. Various extracts from the stem bark were evaluated for anti-parasitic and anti-amoebic effects against *Caenorhabditis elegans* and *Entamoeba histolytica*, respectively. Both ethanol and water extracts did not reveal any notable inhibition of both *C. elegans* and *E. histolytica*. These results may well suggest that the anti-parasitic compounds are embedded within the leaves of the plant species, as reported by Shai et al. [103].

### 5.6. Wound Healing Effect

The wound healing effect of extracts (aqueous and acetone) and isolated compounds (lupeol and betulinic acid) on female Wistar rats weighing 150 to 200 g was evaluated using aqueous cream as a negative control and amphotericin as a positive control [47]. Compounds were dissolved in aqueous cream and tested at 1 and 2%, while extracts were tested at 10 and 20%. The rats were wounded and infected with some colonies of *Candida albicans*. Later, various extracts and compounds at different concentrations were applied to the wound, and the healing effect was measured using parameters such as erythema, exudates, crust formation, and lesion size. The exudate release on wounds was less in wounds treated with 2% betulinic acid than in cream-treated controls, while treatment with 1% and 2% lupeol resulted in the highest exudate formation. Further, the treatment with a 20% water extract formulation resulted in higher exudate release than the control wounds treated with cream only, while treatment with amphotericin B inhibited exudate release from wounds. In short, the 2% betulinic exhibited a better wound healing effect compared to lupeol, cream, and extracts from *C. dendata*. After 13 days of treatment, the rats showed no infections with the fungal strain. These may well suggest that the extracts and isolated compounds exhibit antimicrobial activity both in vitro and in vivo. On wound size measurements, the acetone extracts exhibited higher wound sizes, hence a lower and poor wound healing effect. Generally, as days progressed, the wound sizes were reduced in most treatments. Pronounced erythema was observed on both 10 and 20% acetone extracts, which may imply that the rats had some inflammation. The results suggest that betulinic acid isolated from the plant species plays a major role in wound healing.

### 5.7. Anti-Diabetic and Anti-Obesity Activity

*Curtisia dendata* demonstrated a notable anti-diabetic potential in vitro. The acetone extract of the leaf exhibited 63.72% glucose utilization activity in 3T3-L1 adipocytes at the highest concentration of 500 µg/mL [38]. The extract further yielded the glucose utilization activity of the C_2_Cl_2_ muscle cells (62.69%) at the highest tested concentration of 500 µg/mL. Although it is difficult to compare these results with the other literature data because the IC_50_ was not extrapolated from the raw data, one can suggest that the extract exhibited a mild to moderate anti-diabetic effect as the IC_50_ in both assays could be estimated just below 500 µg/mL. However, it is important to notice that there is a general relationship between diabetes and obesity. Children with obesity challenges are highly likely to develop both types of diabetes sometime during their growth, particularly after the teen stage [105,106]. Some authors have also proposed that the plant species that can exhibit a potential anti-diabetic potential are highly likely to curb obesity in adults [25].

## 6. Toxicology

Although traditional medicine is used to treat devastating human and animal infections, the safety profiles of such traditional medicines may be over g/mL or less, as the focus is on the therapeutic potential of the plant species, particularly against multi-resistant pathogenic strains [107]. The 70% methanol extract from the stem bark was evaluated for toxicity using the Brine Shrimp Lethality (BSL) assay [40]. The extract exhibited an LC_50_ value of 320 µg/mL against *Artemia salina*, while the standards, cyclophosphamide and vincristine phosphate, exhibited LC_50_ values of 16.3 and 0.52 µg/mL, respectively. These results suggest that the extract may not be toxic to *A. salina*. However, it is important to notice that the results of the BSL assay do not translate into the mammalian cell studies. Furthermore, the drug vehicle, which is often a solvent mostly used in extracting the solvent, gives false positive results due to the toxicity of the solvent, not the drug or extract itself [108]. The in vitro cytotoxicity studies of *Curtisia dendata* extracts and isolated compounds from the plant species have been against HEK293 and Vero cells [43], while the in vivo studies were performed in Wistar rats [64]. Lupeol was less toxic compared to betulinic acid, with LC_50_ values of 89.5 and 10.9 µg/mL against Vero cells, respectively. It is important to notice that the cytotoxicity of betulinic acid was comparable to that induced by the positive control, berberine, with an LC_50_ value of 10 µg/mL [102].

In addition to ursolic acid, which exerted some degree of cytotoxicity against the HEK293 cell line (LC_50_ = 122.4 µg/mL), betulinic acid, lupeol, and β-sitosterol exhibited no cytotoxic effect against similar cell lines, yielding an LC_50_ value of >300 µg/mL [43]. Although betulinic acid exerted a notable cytotoxic effect against Vero cells, it is important to note that the in vitro results do not always equate to the in vivo studies [109]. Furthermore, other authors report that the plant species needs to possess a LC_50_ value of 100 µg/mL or less to be referred to as cytotoxic to normal cell lines [110,111]. The cytotoxic effect of the extracts, fractions, and isolated compounds from the plant species still needs to be explored, as there are only a few reports in the literature for comparison purposes and a possible further in vivo study. The toxicity of an aqueous extract from *Curtisia dendata* stem bark has been evaluated against healthy male Wistar rats [64]. The rats were given an oral dose of the extract at 50, 100, and 200 mg/kg once per day for a period of 28 consecutive days. The extract did not show any signs of toxicity to the rats, and there was no significant difference between the treatments and controls. These results may well suggest that the plant species have no toxicity in vivo at the highest concentration of 200 mg/kg and could well be used in traditional medicine to treat a variety of infections.

## 7. Other Important Uses of *Curtisia dendata*

Although the plant species is traditionally used in the treatment of many human and animal infections [9], it is important to note that there are other important economic uses. For example, the wood from *Curtisia dendata* has long been used for the manufacturing of indigenous furniture, heavy flooring, wagons, spokes, musical instruments, and tool handles mostly used in farming [112]. The wood is carved to make an African plate used for carrying food commonly known as “*telo*” amongst the Pedi cultures of South Africa. These utensils could be well carved and sold in traditional markets, generating household income for various communities. The fruits are edible and could be collected in bulk and sold to generate income [113].

## 8. Conclusions

*Curtisia dendata* is used in the treatment of many human and animal infections in Southern Africa. Due to its threatened conservation status, the plant species is found mostly in conserved areas and parks. These have resulted in many authors working on both the biological and phytochemistry of the leaves, which are the only parts likely to be harvested and donated to nature reserves. The in vitro studies revealed that the crude extracts and isolated compounds exhibited noteworthy antimicrobial activity against microorganisms that include *Microsporum canis*, *Sporothrix schenkii*, *Candida albicans*, *Cryptococcus neoformans*, *Mycobacterium tuberculosis*, *Escherichia coli*, *Bacillus cereus*, *Bacillus subtilis*, *Shigella sonnei*, *Shigella typhimurium*, *Streptococcus pyogenes*, *Enterococcus faecalis*, *Pseudomonas aeruginosa*, *Mycoplasma hominis*, *Staphylococcus aureus,* and *Acinetobacter lwoffii*. These important microbes have been implicated as major causative agents of many infections, including skin and related infections. Other microbes are implicated as causative agents of diarrhea and dysentery, sexually transmitted infections, and many opportunistic infections associated with HIV/AIDS. Phytochemically, the plant species exhibited the presence of many classes of compounds, dominated by pentacyclic triterpenes that may well explain the biological activity reported in the current work, hence being used in African traditional medicine to treat a variety of human and animal infections. The plant species further exhibited potential antioxidant, anti-cancer, anti-parasitic, and wound-healing effects. These validate the use of plant species in the treatment of major human and animal illnesses. However, the toxicological parameters, antimicrobial activity against pathogens belonging to the traditional sphere of sexually transmitted infections, anti-diabetic and anti-amoebic effects, anti-mycobacterial effects, and anti-inflammatory activities need to be explored. It is also important to notice that in vitro studies do not always transfer to in vivo studies. Therefore, it is essential to carry out in vivo studies relevant to the plant use of the selected species.

## Figures and Tables

**Figure 1 life-13-02159-f001:**
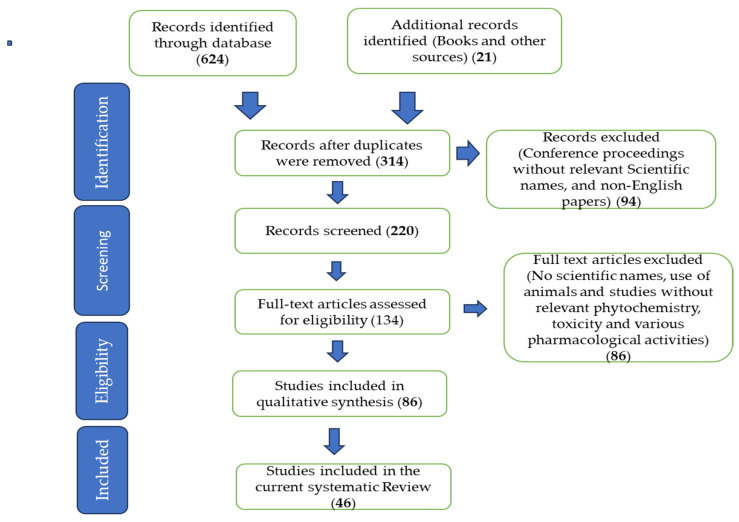
Various sources were extracted from the literature to collate the information reported in the current systematic review.

**Figure 2 life-13-02159-f002:**
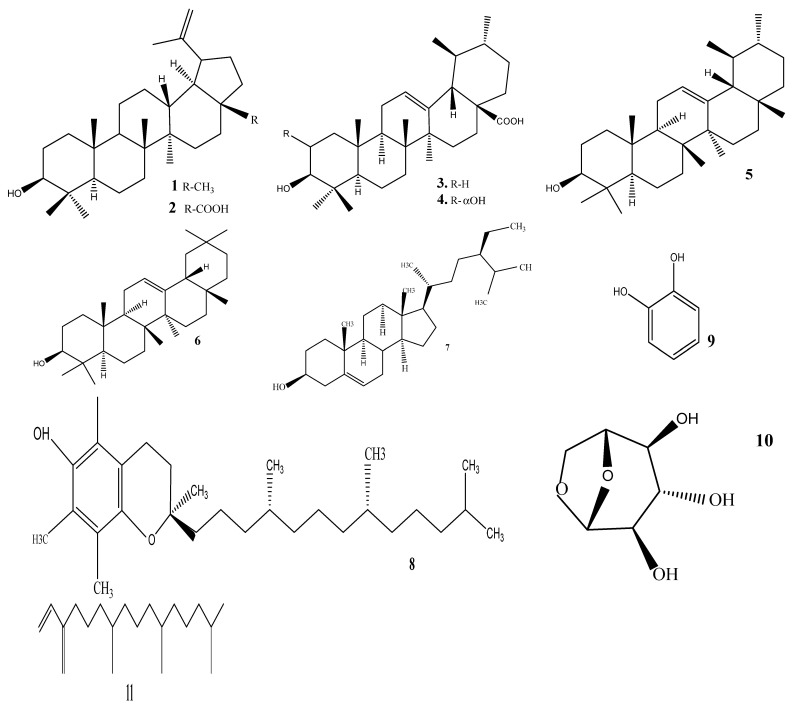
Structures of the identified phytochemicals from *Curtisia dentata*.

## Data Availability

The data presented in this study are available on request.

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
