# Peer review of "A Systematic Review of Curtisia dentata Endemic to South Africa: Phytochemistry, Pharmacology, and Toxicology"

_life, 2023, doi:10.3390/life13112159_

Round 1
Reviewer 1 Report
Comments and Suggestions for Authors
The authors tried to compile the available data about phytochemistry, pharmacology, and toxicology of Curtisia dentata, a plant species in African traditional medicine.
I have a few recommendations:
1. Change the title to "A systematic review of Curtisia dentata endemic to South Africa: Phytochemistry, pharmacology, and toxicology."
2. Move the section "Phytochemistry" right after "Taxonomy, botany, conservation status and distribution of C. dentata."
3. Combine figures 2 and 3. A possible title of the new figure could be "Structures of the identified phytochemicals from Curtisia dentata". Place the number of each structure right below it. Cite the figure in the text.
4. Please, discuss the main findings of the reference "Van Wyk, A. S., & Prinsloo, G. (2021). NMR-based metabolomic analysis of the seasonal and regional variability of phytochemical compounds in Curtisia dentata stem bark. Biochemical Systematics and Ecology, 94, 104197" in section Phytochemistry.
6. Five the unique number of each phytochemical in bold without brackets.
Author Response
Good day.
Please find the attached for your perusal. We have amended the manuscript as required by the Reviewer. Hope all is well. Thank you again for your esteemed comments and support.

Reviewer 2 Report
Comments and Suggestions for Authors
Please, refer to PDF file

Author Response
Good day
The authors are thankful for your comments and have amended the manuscript as required by you as a Reviewer. We truly appreciate your effort. Thank you.

Reviewer 3 Report
Comments and Suggestions for Authors
The reviewed manuscript presents a systematic review and summary of the pharmacological properties and medicinal potential of the monotypic species Curtisia dentata (Burm.f.) C.A.Sm. (Curtisiaceae) which is naturally occurring in Africa. The available scientific data have been properly compiled, although due to the pattern of research on herbal medicinal substances, I suggest a change in the order of chapters 4 and 5 (after the taxonomy, it would be appropriate to present the chemical composition and then the biological/pharmacological properties and toxicology in relation to the safety of use). Authors should carefully proofread and correct minor editorial and punctuation errors (e.g., periods after paper titles or chapter titles). It appears that lines 392 to 406 should be included in section 4.5 Anti-parasitic and anti-amoebic activity instead of 4.7 Anti-diabetic and anti-obesity activity. In the text of the work, there are also spelling mistakes, such as "botulin" instead of "betulin" (line 425), or terms that are not entirely clear, such as "anti-pesticide" (line 40). Beta-sitosterol and other sitosterols (line 420) are plant sterols or phytosterols (modified triterpenes), please take this into account when describing the chemical composition.
Author Response
Good day
We are thankful for the esteemed comments from you as the Reviewer of our manuscript. We are humbled by your dedication and support. Hence, we as authors, have amended the manuscript as per your comments and recommendations. We thank you

Reviewer 4 Report
Comments and Suggestions for Authors
Comments to the Authors,
The manuscript entitled “A systematic review of Curtisia dentata endemic to South Africa: Phytochemistry, toxicology, and important pharmacological activities” has been reviewed. The topic is quite innovative and interesting and the MS is acceptable for publication in the present form.
Concerning the review, there is evidence for a critical assessment of the data, and it remains clear to the reader what the really salient findings are. The manuscript presents a critical analysis of the data and do not simply provide a long list of facts. It should be interesting mentioned the number of selected articles in the abstract, as a result of the activities.
Conclusions are quite complete and it is clear the suggestion about additional studies on composition and biological characteristics.
Author Response
Good day
We are happy that our manuscript fits well in the journal and was well written. We are now resubmitting for final decision. We thank you for your efforts.
